# Effect of Melatonin in Broccoli Postharvest and Possible Melatonin Ingestion Level

**DOI:** 10.3390/plants11152000

**Published:** 2022-07-31

**Authors:** Antonio Cano, Manuela Giraldo-Acosta, Sara García-Sánchez, Josefa Hernández-Ruiz, Marino B. Arnao

**Affiliations:** Phytohormones and Plant Development Lab, Department of Plant Biology (Plant Physiology), University of Murcia, 30100 Murcia, Spain; aclario@um.es (A.C.); manuela.giraldoa@um.es (M.G.-A.); sara.garcias@um.es (S.G.-S.); jhruiz@um.es (J.H.-R.)

**Keywords:** broccoli, human nutrition, improved health, melatonin, postharvest, vegetables

## Abstract

The post-harvest stage of broccoli production requires cold storage to obtain enough days of shelf life. It has been proved that melatonin is useful as a post-harvest agent in fruits and vegetables, including broccoli. In this study, the broccoli heads treated with melatonin have a longer shelf life than the control samples, which was reflected in parameters such as fresh weight, hue angle (expresses color quality), and chlorophyll and carotenoid contents. Treatments with 100 μM melatonin for 15 or 30 min seem to be the most appropriate, extending the broccoli’s shelf life to almost 42 days, when it is normally around 4 weeks. In addition, a study on the possible impact that melatonin treatments in broccoli could have on melatonin intake in humans is presented. The levels of superficial melatonin, called washing or residual melatonin, are measured, showing the possible incidence in estimated blood melatonin levels. Our results suggest that post-harvest treatments with melatonin do not have to be a handicap from a nutritional point of view, but more research is needed.

## 1. Introduction

Increasing the consumption of fruit and vegetables is a key component in a healthy diet to reduce diseases [1,2]. The cruciferous vegetables have shown an important inverse correlation between their intake and oncogenic and cardiovascular diseases [3,4]. Broccoli is a common component of the human diet that has recently seen a growth in demand and increased consumption due to its high nutritional values. Broccoli is easily perishable during the post-harvest period, and its visual and sensory quality is significantly reduced, the loss of sepal greenness accompanied by yellowing being the most visible sign of deterioration. This phenomenon is a major limitation to the post-harvest storage and transportation of broccoli [5]. The green/blue color is the main important commercial quality index in broccoli, due to chlorophyll degradation which is the first visible symptom of senescence [6].

Melatonin (*N*-acetyl-5-methoxytryptamine) is an indolic compound derived from tryptophan discovered in cows [7,8], which plays a role as a hormone in vertebrates. Numerous roles have been proposed for melatonin in mammals, such as the regulator of sleep cycles [9], sexual behavior [10], endocrine rhythms [11,12,13], amongst others. More recently, its implications have been shown in glucose metabolism and insulin [14,15,16], as a sensitizer in anti-oncogenic therapies [17], its positive effects in Parkinson’s and Alzheimer’s diseases [18], and its therapeutic efficiency in COVID-19 treatment [19,20,21].

In plants, melatonin (phytomelatonin) was identified in 1995 [22,23,24]. Since then, a variety of studies have been carried out to understand the role that melatonin plays in plant physiology [25,26,27]. Practically all of the responses in plants have been shown to be modulated by melatonin, which improves processes such as germination, growth, flowering, etc. [28,29,30,31], but the most important action of melatonin is as a mediator in stress situations [32,33,34,35,36].

There is evidence that suggests an important role for melatonin in the regulation of both the biochemical and physiological aspects of postharvest [28,30,37,38]. Concerning broccoli postharvest, melatonin treatment could be an effective technique to improve the quality of fresh-cut broccoli during cold storage [5,39,40,41,42,43]. A recent review on the role of melatonin in broccoli and other Brassicaceae can be consulted [44].

This paper presents a study on the ability of melatonin to improve the conservation of broccoli, both at room temperature and under cold commercial conditions. The changes in weight loss, color and chlorophyll, and carotenoid contents were studied, and, as a novelty, the contents of melatonin (endogenous and exogenous) produced in treated broccoli and its possible impact on human health.

## 2. Material and Methods

### 2.1. Chemicals

The solvents (ethanol, acetone, acetonitrile, and ethyl acetate) and reagents used were from Sigma-Aldrich Co. (Madrid, Spain). Milli-Q system (Milli-Q Corp, Merck KGaA, Darmstadt, Germany) ultra-pure water was used.

### 2.2. Plant Material

The broccoli (*Brassica oleracea* L. var. *italica*, cv. *Parthenon*) was harvested at commercial maturity by a horticultural company in Lorca (37°40′16.28″ N; 1°42′6.12″ W) from the region of Murcia (Murcia, Spain). The harvested samples were placed in polystyrene boxes with ice to avoid bruising and hold moisture. The broccoli heads were transported to our laboratory where pieces of uniform size and color and without apparent disease or injuries were randomly selected for the experiments or treatments.

### 2.3. Broccoli Head Treatments

The broccoli heads were immersed in 0 (distilled water as the control for 30 min), 50 and 100 μM melatonin water-solutions for 15 min and 30 min. The heads were then removed and air-dried for 30 min. All of the procedures were performed at room temperature. Afterward, each head was wrapped in transparent plastic film and stored at 5 °C and 75% relative humidity for 42 days in the dark. In addition, a preliminary study at 20 °C for 7 days with broccoli heads and florets was completed. Each treatment was made three times. The sampling was performed every 7 days to determine weight loss, color, chlorophyll, carotenoid, and phytomelatonin content. The broccoli florets were obtained from heads, and they were immediately frozen in liquid nitrogen and stored at −80 °C for later analysis of chlorophyll, carotenoid, and melatonin contents.

### 2.4. Weight Loss

The weight loss of the broccoli heads was calculated as the percentage change relative to the initial weight using the following formula: weight loss (%) = (IW − FW)/IW × 100; where IW is the initial weight and FW is the weight on a certain day after storage.

### 2.5. Color

The color parameters of the broccoli heads were measured at four points over the surface of each piece with an automatic colorimeter PCE-XXM 20, (PCE Instruments, Tobarra, Spain), calibrated following the instructions. The values determined were L* (changes in L* indicate lightness of plant tissues), and the hue angle (main color quality marker), which was calculated as H = 180° + tan^−1^ (b*/a*) when a* < 0 and b* > 0 [45].

### 2.6. Determination of Chlorophyll and Carotenoid Contents

The changes in the chlorophyll and carotenoid levels were determined according to Lichtenthaler (1987). Briefly, 0.3 g of the frozen powdered broccoli florets (the most superficial tissue of the broccoli heads) were placed in a glass tube with absolute ethanol and then incubated at 100 °C for 30 min (until the tissues were colorless). During the extraction, hot ethanol was refilled when needed. The final volume of extraction was adjusted to 10 mL. The absorbance of the supernatant was measured at 470, 649, and 665 nm to determine chlorophyll a, b, and total, and total carotenoid contents. All of the procedures were performed under diffused light.

### 2.7. Determination of Residual and Endogenous Melatonin

In order to know how the exogenously applied melatonin is distributed in the broccoli heads during the different treatments, the content of the residual melatonin on the surface and the content of the melatonin absorbed by the tissue (endogenous melatonin) were determined. Three broccoli heads were treated with 0 (control) and 100 µM melatonin at two immersion times (15 and 30 min). After drying in the air, the entire surface of the broccoli head was washed homogeneously with 200 mL of ethanol:water (50:50) and drained for 10 min. The final volume collected (ethanol:water washes) was determined and kept in the dark at 5 °C until the analysis of the melatonin content. In addition, after the ethanol:water washes, all of the florets of each broccoli head were crushed to determine the content of endogenous melatonin. The determination of the residual and endogenous melatonin content was carried out by LC with fluorescence detection.

The melatonin was measured by liquid chromatography with fluorescence detection (LC-FLUO) [46]. Briefly, the crushed florets (0.2 g) were mixed with ethyl acetate (4 mL) and shaken overnight (15 h) in the dark. The samples were filtered, and the solvent was evaporated to dryness under vacuum using a SpeedVac (ThermoSavant SPD11V, Thermo-Fisher Sci, Waltham, MA, USA) coupled to a refrigerated RCT400 vapor trap. The dry residue was redissolved in acetonitrile (1 mL), filtered (0.2 µm), and analyzed. A Jasco liquid chromatograph Serie-2000 (Tokyo, Japan) equipped with an online degasser, quaternary pump, autosampler, thermostatted column, and a Jasco FP-2020-Plus fluorescence detector were used to analyze the melatonin content. A Waters XBridge C18-S5 column (2.1 mm × 100 mm) thermostatized at 36 °C was used. The mobile phase consisted of water:acetonitrile (80:20) at an isocratic flow rate of 0.5 mL/min. The fluorescence detector was programmed with an excitation value of 280 nm and 350 nm of emission. The data were analyzed using the JascoChromNAV v.1.09.03 Data System Software (Tokyo, Japan). The melatonin identification was carried out by comparing the excitation and emission spectra of standard melatonin with the corresponding peak of melatonin in the samples. The melatonin quantification was determined using a standard curve and the data were expressed as ng residual melatonin/broccoli head or as ng melatonin/g FW.

### 2.8. Graphic and Statistical Analysis

The graphical analysis of the data was made using SigmaPlot program version 14 (SYSTAT Software Inc., San José, CA, USA). Analysis of variance was performed using IBM SPSS Statistics 22.0 (IBM, New York, NY, USA). The statistical significance was considered for *p*-values less than 0.05 in ANOVA and a post-hoc with the Tukey HSD test.

## 3. Results and Discussion

### 3.1. Effect on Color Parameters of Broccoli Intact (Heads) and Florets Treated with Melatonin and Stored at 20 °C

A preliminary study was carried out on the broccoli heads (intact) and florets (about 7 cm high and 28 g weight) at a storage temperature of 20 °C, since at this temperature the physiological changes occur more quickly. The following melatonin concentrations were applied: 0 (control), 50, 100, 200, and 500 µM in the broccoli florets, and in the broccoli heads at 0, 50, and 100 µM melatonin, and at a fixed immersion time of 30 min. In all of the cases, they were left to air dry and wrapped in a plastic film. They were left in a culture chamber, at a controlled temperature in the dark; the color was measured before treatment and after 7 days of storage.

The results obtained in the color measurements on the broccoli florets stored at 20 °C are shown in Figure 1. The initial value of L* on day 0 gave a mean value of 47.8. After 7 days, a slight increase is observed for all of the concentrations of the melatonin-tested broccoli, but the highest increase is observed in the control (L* = 54.0); this turns out to be 6.95% higher (1.07 times) compared to the mean of the L* values of all of the broccoli florets treated with melatonin (L* = 50.48) (Figure 1A). The L* parameter indicates the brightness or luminosity of the broccoli surface. Generally, it is desirable that it remains constant.

The initial reading of the mean value of the hue angle (H) was 137.02 for all of the florets analyzed, observing a general decrease after 7 days (Figure 1B). The hue angle indicates the color changes, according to the equation presented in Section 2, using as a reference the color bar that appears in Figure 1. These results indicated that the green/blue color of the florets was lost at 20 °C after 7 days of storage. However, when analyzed by treatment on 7th day, it is observed that 100 µM MEL preserved a better green/blue color (H = 134.67) and followed by 200 µM MEL (H = 125.94) relative to the control (H = 124.24) after 7 days. The largest drops in the hue value were observed for the MEL 50 and 500 µM treatments (H value was 112.50 and 116.75, respectively).

In the intact broccoli heads, the changes in the color parameters in the control and melatonin treatments are shown in Figure 2. The initial value of L* on day 0 was 49.62. After 7 days, a significant increase for 50 µM MEL (L* = 59.21) concerning the control (L = 51.62) was measured, and a practically maintained value in 100 µM of MEL (L = 49.77) (Figure 2A).

Regarding hue angle (H) in the broccoli heads (Figure 2B), a general decrease was observed after 7 days (control, H = 119.70 and 50 µM MEL, H = 120.46) vs. day 0 (H = 138.49), with a very slight increase for 100 µM MEL (H = 135.15) in respect to the control of day 7. The H data pointed to better maintenance of the color after 7 days at 20 °C for 100 µM MEL.

### 3.2. Effect on Weight Loss, Color, and Photosynthetic Pigment Contents of Broccoli Heads Treated with Melatonin over Storage Time at 5 °C

Based on the preliminary results obtained at 20 °C and 30 min of immersion with melatonin, it was decided to carry out a study of the effects of cold storage (5 °C) in melatonin-treated broccoli. For this purpose, 100 µM MEL was selected since, at 20 °C, it produced a significant delay in the loss of color compared to the control and the other melatonin concentrations applied (Figure 1 and Figure 2). On the other hand, it was proposed to study the relevance of the immersion time, and to observe if a time less than 30 min could improve the melatonin effect, because timing can be an important parameter in the vegetable packaging industry.

The weight loss of the broccoli heads during storage at 5 °C were determined, because water loss is one of the main issues during broccoli storage, causing stalk hardening and bud-cluster turgidity loss [47,48]. As can be seen in Figure 3, the weight loss was always greater in the control samples than in the melatonin-treated samples, throughout all of the measured days. Regarding the different melatonin treatments, it was observed that melatonin, in all of the cases tested, acted by delaying water loss, which is related to its participation in the closing/opening of the stomata [49]. Of all of the treatments, the 100 µM MEL applied for 15 min was the most effective, since it reduced water loss by 6% compared to the control, while the rest of the treatments did not show significant differences between them (Figure 3).

Variations in the visual quality during the broccoli yellowing were measured through the parameters L* and hue angle (H) (Figure 4). The L* values in both the control and the melatonin-treated broccoli heads increased gradually during storage at 5 °C. The melatonin-treated heads showed a slightly lower rate of increase in the L* value than the control heads, at 28, 35, and 42 days, being statistically similar for all of the melatonin concentrations and immersion times tested (Figure 4A).

The initial H value was 132.13 (day 0), remaining more or less constant during the 5 °C storage until days 35 and 42, where the control presented a significant drop, reaching an H value of 124.59. However, the treatments with melatonin for both of the immersion times (15 and 30 min) delayed this drop, with a H value of 132.78 (Figure 4B), in 35 days. After 42 days, only the melatonin treatments of 15 min could maintain the color, showing a mean H value of 133.27. These results indicate that the color of the heads, stored at 5 °C, was gradually lost and that the melatonin delays it; this indicated that a treatment with melatonin would increase the shelf life of the broccoli beyond 21 days, and may even reach 42 days of storage with acceptable color.

In order to search for a correlation between the color parameters and chlorophyll and carotenoid contents, these were determined in the melatonin-treated broccoli throughout the cold storage time. A net symptom of senescence in the broccoli heads is the loss of green/blue color or yellowing mainly due to the catabolism of chlorophylls. Based on the results obtained from the evaluation of the color (Figure 4), the pigment contents in the 100 µM melatonin-treated samples for 15 and 30 min were determined.

The total chlorophyll content showed a decrease in both the treated and the control samples, indicating postharvest senescence (Figure 5A). However, in the case of the melatonin-treated heads, the loss of chlorophyll appears clearly attenuated compared to the control. Moreover, especially from day 28 to the last day of storage, the total chlorophyll values were higher than those of the control. In addition, on day 42 of storage, the 15 min immersion treatment of 100 µM MEL resulted in a total chlorophyll value of 0.42 mg/g FW, being 0.23 mg/g FW for the control, that is, approximately twice the chlorophyll content.

Similarly, the total carotenoid contents remain constant for up to 14 days (Figure 5B). After 21 days of storage, an increase in the carotenoid levels was observed, with a different behavior in the control and the melatonin-treated broccoli heads. In general, and after a significant carotenoid content increase after 21 days, the values in the control remained constant until the last day of the study. In contrast, the melatonin-treated broccoli showed an increase in the carotenoid levels, reaching up to 40% more than the control. Regarding the treatment times tested, higher carotenoid values were obtained on day 42 for the 30-min treatment.

There are several actions to be applied to achieve a longer shelf life of broccoli, preserving its excellent nutritional qualities due to its high contents of several vitamins, highlighting pro-vitamin A, B3, and C, and other substances such as lutein, quercetin, folic acid, glucosinolates, and fiber [50]. The coating of the broccoli in plastic film, the packaging in a tray, or its distribution in bulk are processes that all must go through an adequate cold chain. In some cases, the ethylene absorbers and modified atmospheres have been used with acceptable results [47]. The use of alcohol vapors has also been proposed as an alternative [51]. The artificial preservatives have been used, with poor results. Lately, the use of natural compounds is an ideal strategy, given the EU regulatory constraints. The use of phytoregulators, such as cytokinins, jasmonates, polyamines, and others, has had very variable results [52,53]. In some cases, the natural preservatives, such as essential oils, have been used, which due to their antimicrobial activity can have some beneficial effects if thermal control is not rigorous [54]. However, the essential oils have the downside of providing undesirable odors and flavors to the fresh product. In addition, generally the cost–benefit balance must be considered, since in most of the cases the treatments are expensive.

Today, the product is usually transported refrigerated in polystyrene boxes containing the heads of broccoli with ice; all of this to ensure a temperature as close to 4 °C, where the broccoli usually maintains its organoleptic qualities [55]. The refrigerated transport is usually carried out in days, achieving good results since the product can have an optimal commercial life of around 25 days. Other studies have suggested that transport conditions of total darkness do not favor its conservation, suggesting that small light treatments can extend the half-life of the product and improve its visual and organoleptic qualities [48]. In other cases, the vibrations due to transport have been shown to cause significant deterioration and a decrease in quality and useful life.

As usual, the temperature is a decisive factor in ensuring the stability in color, appearance, and nutrient contents of broccoli. Thus, a clear difference can be found in conditions of 20 °C and 5 °C for each of the estimated parameters. Generally, the shelf life of the broccoli preserved at 20 °C is about 3–4 days, and can be extended up to almost 5–8 days with melatonin treatments [5,39]. However, in our assays after 7 days at 20 °C, both the luminosity (L*) and the color index (H) were quite well maintained (without increasing L* too much and without decreasing H) in the case of the broccoli treated with melatonin (Figure 1 and Figure 2). For other authors, in these conditions, the melatonin treatments significantly prevented the increase in the L* value and delayed the decrease in the H index [5,39,40].

When cold storage and melatonin treatments were combined, a significant improvement in the broccoli quality can be determined, even at extended shelf life. At 5 °C, the melatonin-treated broccoli heads with 100 μM melatonin preserved considerably the weight loss in the period 7–42 days (Figure 3), indicating that melatonin can regulate open/closed stomata in the broccoli heads as it occurs in the leaves, saving water losses [28,49,56]. Concerning the values of L* and H, the melatonin treatments slightly decrease the luminosity of the broccoli heads especially at long times (days 28, 35 and 42) (Figure 4A). The opposite occurs with the color (H index), a significant maintenance of the bluish-green color compared to the respective controls occurs in melatonin treatments on days 35 and 42, (Figure 4B), presenting a significantly less yellowing level.

The chlorophyll and carotenoids are the main items responsible for the broccoli color. Their chlorophyll content losses occur as a habitual senescence process similar to leaves [6]. In the control broccoli heads, the total chlorophyll contents decreased exponentially, reaching a chlorophyll loss of 62% after 42 days of storage at 5 °C (Figure 5A). The melatonin preserved the chlorophyll content in the treated broccoli all of the time, being very significant on days 28, 35, and 42, reaching a chlorophyll content in 42 days similar to the control broccoli after 14 days of conservation, a differential of 4 weeks. This regulatory action of the melatonin on senescence was first described by our group in barley leaves [57]. Subsequently, melatonin was found to have improved the rate of photosynthesis, PSII efficiency, Rubisco activity, the chlorophyll and carotenoid levels, stomatal conductance, and leaf-intercellular CO_2_ contents. The Rubisco small subunit (RbcS) and many chlorophyll *a/b* binding-protein transcripts were upregulated by melatonin. Instead, the senescence transcription factor SAG12 and some chlorophyll degradation factors, such as pheophorbide *a* oxygenase (PaO), were downregulated [58,59].

As far as the carotenoid contents are concerned, there was an increase in their levels that appeared from day 21, due to the senescence process. However, in the case of the broccoli treated with melatonin, a greater increase in the carotenoid biosynthesis can be observed (Figure 5B). This carotenoid promoting effect can be explained because melatonin upregulates the carotenoid biosynthesis genes, thereby increasing the levels of several carotenes, including α-, β-carotene, lutein, zeaxanthin, and lycopene, and upregulating several carotenogenesis genes, such as DXS, DXR, GGPPS, PSY, PDS, ZDS, CRTISO, and CYCB. This action is mediated by ethylene since melatonin also upregulates ethylene biosynthesis enzyme transcripts (ACO and ACS), several ethylene-signal elements (*EIL1, EIL3*, and *ERF2*), and ripening factors (*RIN*, *CNR*, and *NOR)*, and downregulating *AP2a*, [28,60,61]. As a result, melatonin can prolong the shelf life in numerous fruits, including tomato, strawberry, peach, banana, litchi, plum, grape, pear, and mango [30]. This carotenoid-promoting biosynthesis by melatonin has been also described in microalgae [62].

### 3.3. Residual and Endogenous Melatonin Contents: Implications for Human Consumption

As already mentioned, melatonin is a molecule with numerous beneficial functions for humans. Therefore, it is of interest to know the amount of melatonin that we could ingest in our diet, due to vegetable/fruit post-harvest treatments with melatonin. In the present study, the residual melatonin levels appearing in the melatonin-treated broccoli heads were quantified. Thus, after the melatonin treatments, the broccoli heads were washed with ethanol:water (50:50) and these washing solutions were subjected to the quantification of residual melatonin by LC-FLUO. Other ethanol:water ratios, such as 80:20 and 60:40, showed results very similar to those of 50:50. This is logical since melatonin is an amphoteric molecule with ease of solubilization in aqueous and also organic media. Figure 6A shows the residual melatonin quantifications present in the washing solutions by each broccoli head. As can be seen, a similar amount of residual melatonin can be found in all of the treatments carried out (100 µM melatonin-treated samples during 15 and 30 min).

It is also important to know the amount of endogenous melatonin absorbed due to the exogenous treatments. As can be seen in Figure 6B, the values of the endogenous melatonin in the control and the melatonin-treated broccoli after washing were similar, being very low (0.5–2.2 ng/g FW) in all of the cases. In the cases of the broccoli heads treated with melatonin that were not subsequently washed, levels between 150 and 225 ng/g FW were measured (Figure 6B).

Therefore, we can establish that melatonin with the possibility of being ingested is the one that can be removed by washing vegetables, and that we can assess its intake. Figure 7 shows a diagram that helps us to make a quantitative estimation of the possible intake of melatonin present in the treated broccoli. A first possibility is to wash the broccoli before intake. As we have seen in Figure 6B, if the broccoli is washed, all of the superficial melatonin, called washing or residual melatonin, is practically removed. The second possibility would be to eat the melatonin-treated broccoli without washing it. In this case, if we consider a serving size of 76 g (according to FoodData Central of USDA), we could ingest about 570 ng/cup. We consider that approximately 75% of the melatonin that is ingested orally is not absorbed or catabolized in the liver, occurring in feces, mainly through bile. Overall, the pharmacokinetic data of the melatonin pointed to a non-absorption in the gastro-intestinal mucosa and a catabolism of melatonin in the liver around 70–90% of intake, with a half-life of melatonin in blood of 10–30 min [63,64,65,66,67,68]. Thus, only 25% would reach the bloodstream, or about 142 ng of melatonin. Considering that the total volume of blood per person ranges between 4.5 and 6 L, an ingested melatonin level of 24–32 pg/mL can be expected; an amount in the usual range for a middle-aged person (see graph in Figure 7) [69,70,71]. Therefore, in our trials with the broccoli treated with melatonin; its theoretical intake would not alter the usual levels of melatonin in the blood.

In conclusion, the intake and absorption of melatonin after post-harvest treatments should be carefully studied to determine the possible impact on melatonin levels in the diets and subsequently in the blood. Studies so far are controversial, with some pointing to a possible effect of melatonin ingested in fruits on blood levels, while in other cases no such increases are observed [72,73,74,75,76,77,78,79]. In general, there has been criticism on the lack of quantitative correlation between the ingested phytomelatonin and its plasma levels [80]. Therefore, more studies and models are needed on aspects such as the bioavailability of phytomelatonin, its time course of assimilation and degradation, and the possible effects on melatonin release rhythms from the pineal gland [71,81,82].

According to our data, the recommendations for the use of melatonin as a natural agent for post-harvest control of fruits and vegetables point to null or low inference in the usual levels of melatonin in humans, indicating that it does not have to be a handicap from a nutritional point of view, but more research is needed.

## Figures and Tables

**Figure 1 plants-11-02000-f001:**
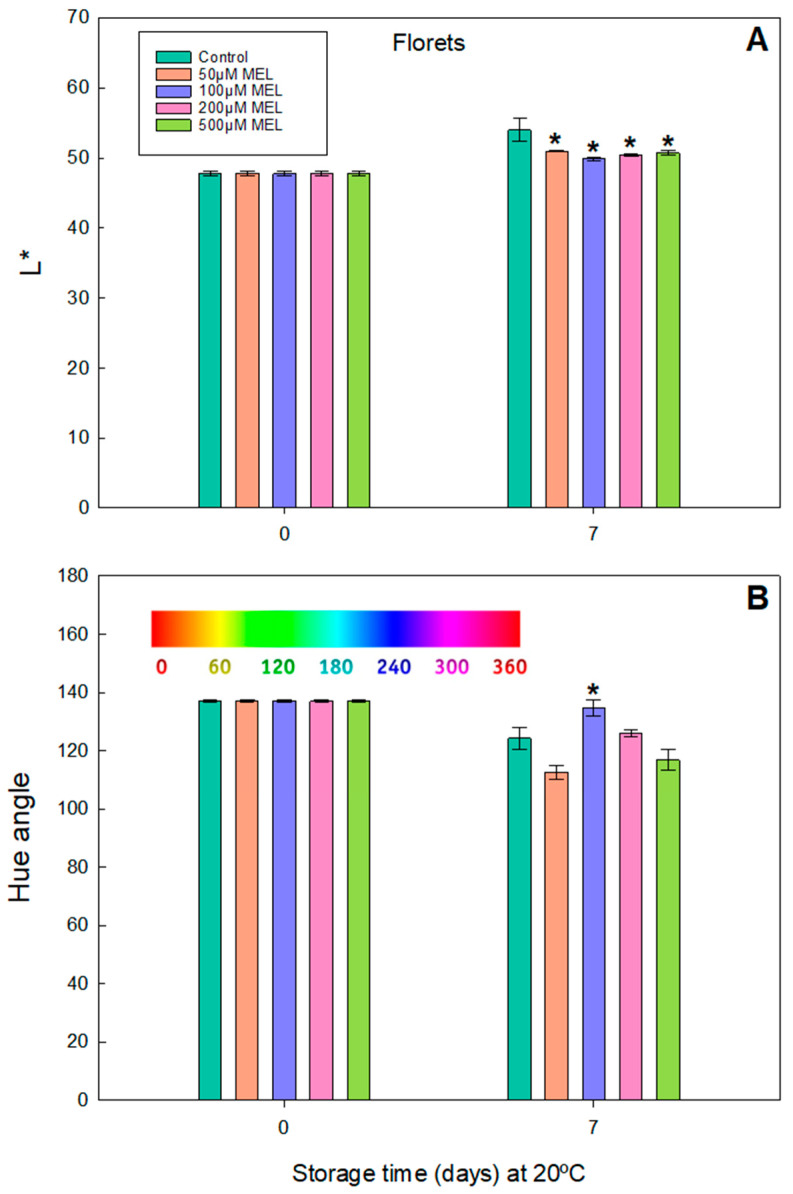
Color parameters in broccoli florets stored at 20 °C. (**A**) Luminosity (L*) and (**B**) Hue angle of broccoli florets treated with 50, 100, 200 and 500 µM of melatonin and stored in the dark at 20 °C for 0 and 7 days. Each bar represents the mean value ± standard error (*n* = 3) obtained for each concentration used. Asterisks indicate significant differences with the control of that day based on a Tukey HSD test with a significance level of *p* < 0.05. In (**B**), the Hue angle scale has been incorporated as a color guide.

**Figure 2 plants-11-02000-f002:**
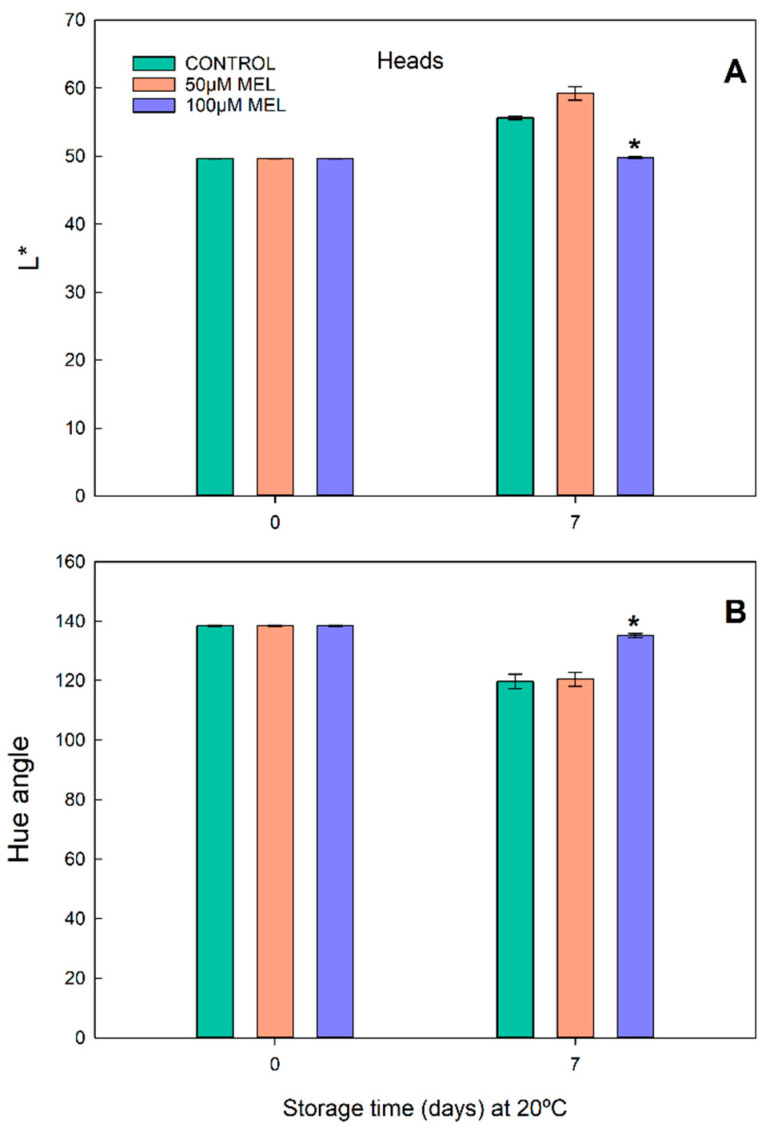
Color parameters in broccoli heads stored at 20 °C. (**A**) Luminosity (L*) and (**B**) Hue angle of broccoli heads treated with 50 and 100 µM of melatonin and stored in the dark at 20 °C for 0 and 7 days. Each bar represents the mean value ± standard error (*n* = 3) obtained for each concentration used. Asterisks indicate significant differences with the control of that day based on a Tukey HSD test with a significance level of *p* < 0.05.

**Figure 3 plants-11-02000-f003:**
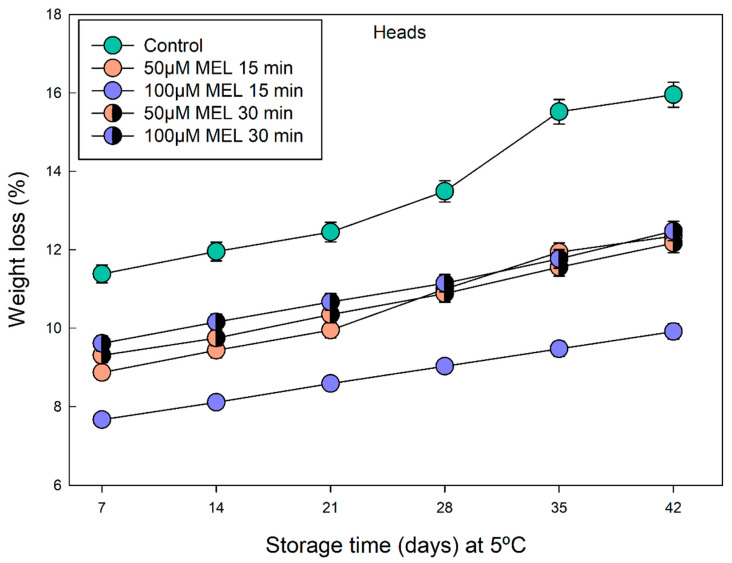
Weight loss of broccoli heads treated with 50 and 100 µM of melatonin during 15 and 30 min and stored in the dark at 5 °C for 0, 7, 14, 21, 28, 35 and 42 days. Each point represents the mean value and ± standard error (*n* = 3) obtained for each concentration used.

**Figure 4 plants-11-02000-f004:**
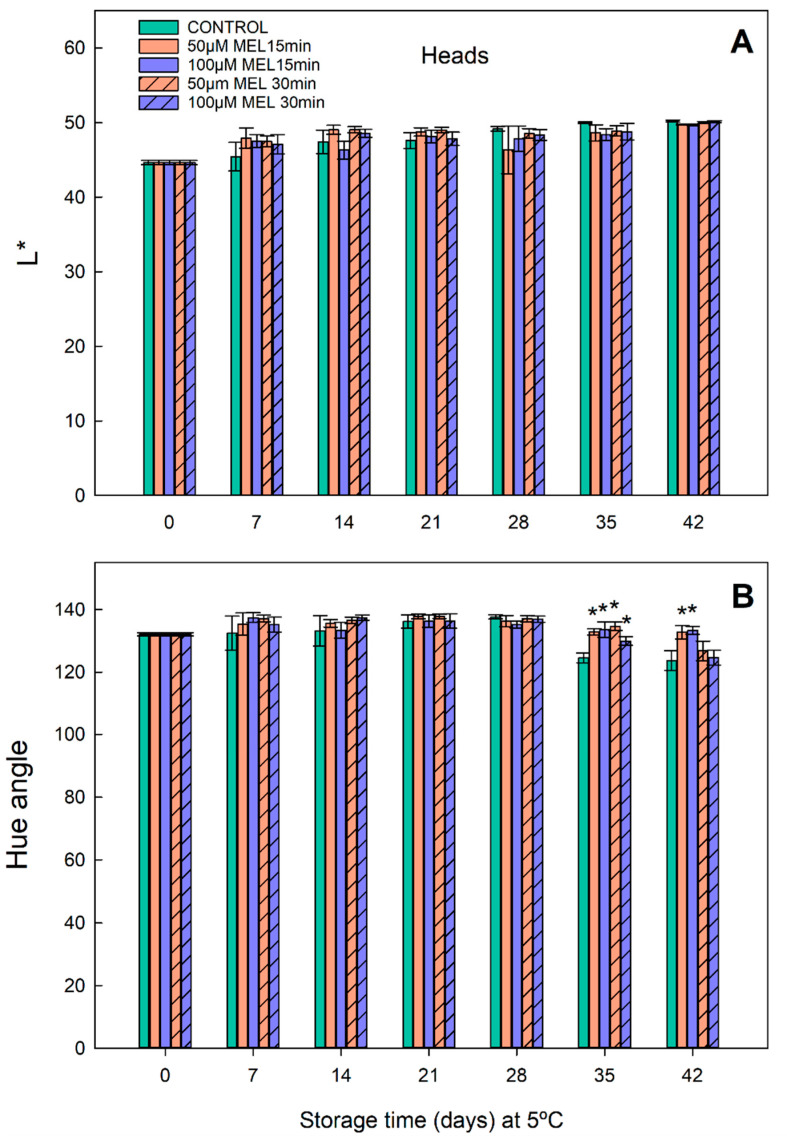
Color parameters in broccoli heads stored at 5 °C. (**A**) Luminosity (L*) and (**B**) Hue angle of broccoli heads treated with 50 and 100 µM of melatonin during 15 and 30 min and stored in the dark, at 5 °C, for 0, 7, 14, 21, 28, 35 and 42 days. Each bar represents the mean value ± standard error (*n* = 3) obtained for each concentration used. Asterisks indicate significant differences with the control of that day based on a Tukey HSD test with a significance level of *p* < 0.05.

**Figure 5 plants-11-02000-f005:**
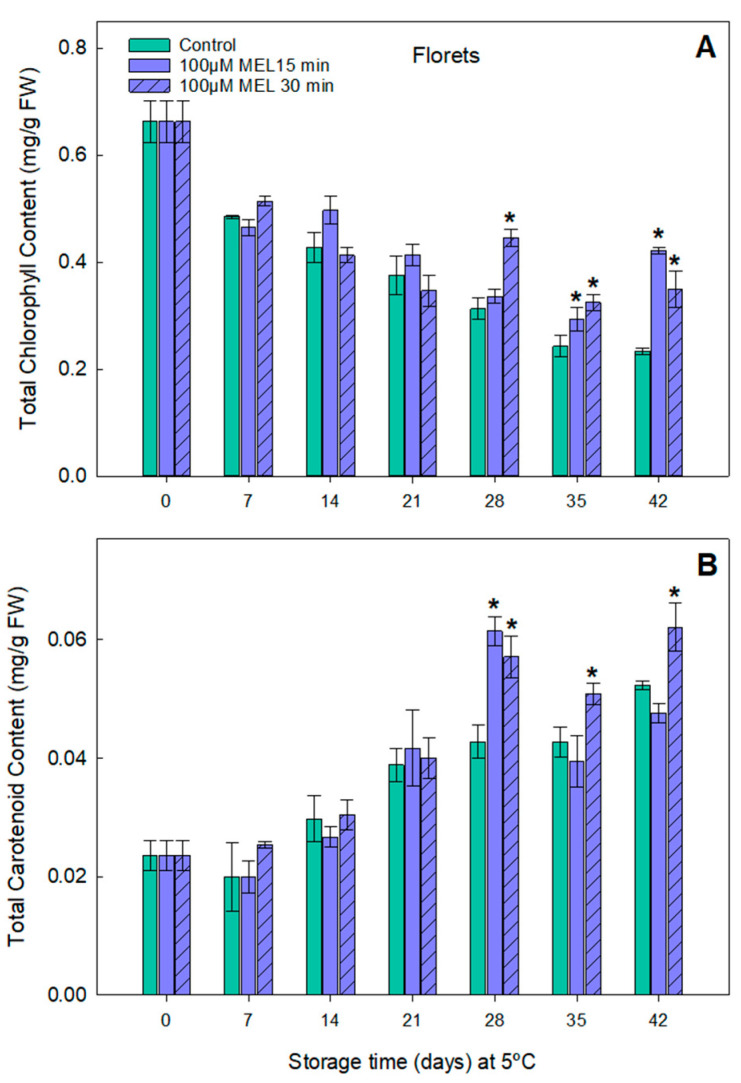
Effect on total content of chlorophylls (**A**) and carotenoids (**B**) in broccoli treated with 100 µM of melatonin during 15 and 30 min and stored in the dark, at 5 °C, for 0, 7, 14, 21, 28, 35 and 42 days. Each bar represents the mean value ± standard error (*n* = 3) obtained for each concentration used. Asterisks indicate significant differences with the control of that day based on a Tukey HSD test with a significance level of *p* < 0.05.

**Figure 6 plants-11-02000-f006:**
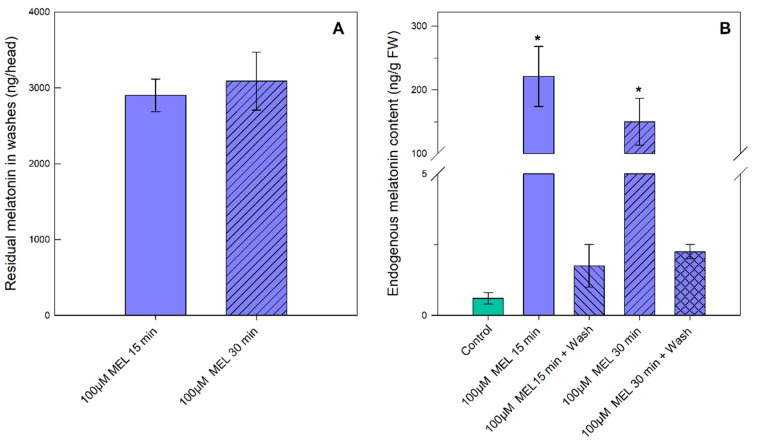
Melatonin quantification by LC-FLUO: (**A**) in the wash solutions of melatonin-treated broccoli heads; (**B**) endogenous melatonin content of diverse broccoli treated samples. Each bar represents the mean value ± standard error (*n* = 3) obtained for each concentration used. Asterisks indicate significant differences between washed and unwashed broccoli heads based on a Tukey HSD test with a significance level of *p* < 0.05.

**Figure 7 plants-11-02000-f007:**
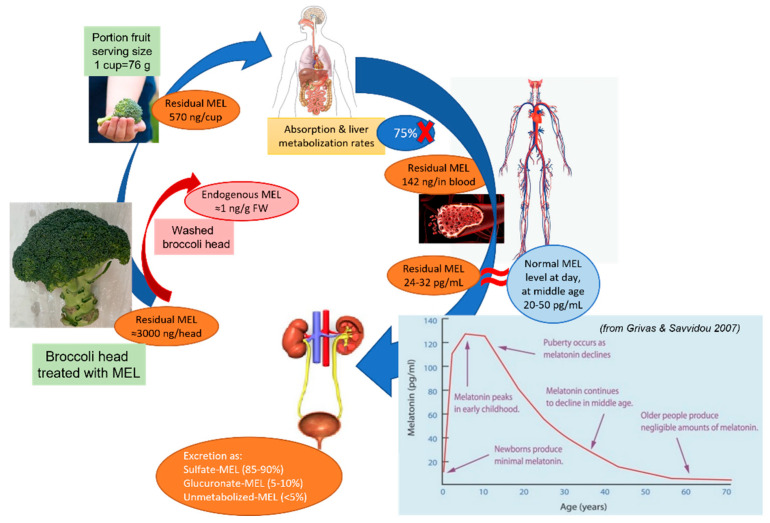
Schematic representation of the exogenous melatonin intake of broccoli treated with melatonin and its theoretical evolution in the human body.

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
