# Peer review of "Effect of Melatonin in Broccoli Postharvest and Possible Melatonin Ingestion Level"

_plants, 2022, doi:10.3390/plants11152000_

Round 1
Reviewer 1 Report
Cano et al. tested the physiology effect of melatonin on post-harvest of broccoli, which is an important vegetable to our daily life. Experiments are designed simple and clear, and results could provide readers scientific ideas on the application of melatonin to vegetables. The main drawbacks are most of the data presented without standard deviation (I will not accept this type of data presentation). Please also include sample number (N) in the figure legends. Another main concern is that you washed broccoli with ethanol:water (please label the percentage) and tested the endogenous melatonin changes. However, the real life won't use ethanol:water to wash broccoli, have you compared the two ways of washing? The last concern is that how do you make sure 75% of take-in melatonin won't be absorbed and won't make any effect to humans?
Minor:
Please provide representative images of broccoli before and after treatment at each time point if possible.
Reviewer 2 Report
This manuscript overall is interesting and is relevant to the readership of plants. Suggestions for improvements have been made on the attached pdf of the manuscript. Greater clarification is needed in the methods with regards to the experimental design and the control, including the context of the control with regard to current industry standard practice. Suggestions are also given to improve the ease of interpretation of some of the figures.
